# A Multimodal Class-Incremental Learning Benchmark for Classification Tasks

## Abstract

Continual learning has made significant progress in addressing catastrophic forgetting in vision and language domains, yet the majority of research has treated these modalities separately. The exploration of multimodal continual learning remains sparse, with a few existing works focused on specific applications like VQA, text-to-vision retrieval, and incremental multi-tasking. These efforts lack a general benchmark to standardize the evaluation of models in multimodal continual learning settings. In this paper, we introduce a novel benchmark for Multimodal Class-Incremental Learning (MCIL), designed specifically for multimodal classification tasks. Our benchmark comprises a curated selection of multimodal datasets tailored to classification challenges. We further adapt a widely used vision-language model to multiple existing continual learning strategies, providing crucial insights into the behavior of vision-language models in incremental classification tasks. This work represents the first comprehensive framework for MCIL, establishing a foundation for future research in multimodal continual learning.

## 1 Introduction

Continual learning aims to develop models that can learn incrementally, integrating new knowledge while retaining the one acquired on previous training iterations. This challenge, while being a representative scenario of the well known stability-plasticity dilemma (Mermillod et al., 2013), has gained considerable traction, particularly in vision or language domains, where significant progress has been made to mitigate catastrophic forgetting. However, despite this progress, the majority of research has treated them separately. This limitation has hindered the exploration and comparison of more complex, pure multimodal scenarios where information from multiple modalities must be processed and retained simultaneously.

Multimodal learning, which combines data from different sources like images, text, or audio, among others, offers the potential for richer representations and enhanced understanding. Yet, applying continual learning to such settings has been underexplored. Most existing works in multimodal continual learning have focused on specific applications, such as Visual Question Answering (VQA) (Qian et al., 2023), text-to-image retrieval (Wang et al., 2021; Sun et al., 2024), and task-incremental settings (Srinivasan et al., 2022), whereas classification tasks lack a standardized benchmark for comparing results and tracking scientific advancements, reducing generalization. This gap has made it difficult to assess progress systematically and fairly across approaches, leaving the challenge of developing new solutions and standardizing multimodal continual learning unresolved.

Several recent works have attempted to address continual learning in isolated modalities by introducing benchmarks for incremental object detection (Han et al., 2021; Verwimp et al., 2023), natural language understanding (Madotto et al., 2020), and other single-modality tasks (Lin et al., 2021). However, extending these frameworks to multimodal settings presents unique challenges due to the added complexity of modality-specific representations and the interactions between them. For example, models must learn how to retain visual features while simultaneously updating language-based understanding—an inherently more challenging scenario than unimodal tasks. Furthermore, the diverse nature of multimodal data complicates the design of incremental learning strategies, which need to balance between modality-specific knowledge retention and cross-modal alignment.

To address these challenges, we introduce a novel Multimodal Class-Incremental Learning (MCIL) benchmark tailored for multimodal classification tasks. Our benchmark includes a curated set of multimodal datasets that span a variety of vision-language classification tasks, providing a comprehensive platform for evaluating the performance of continual learning algorithms in a multimodal setting. To ensure the benchmark's relevance and utility, we adapt the widely used vision-language model Flava (Singh et al., 2022) to multiple existing continual learning strategies, offering insights into how these models behave in incremental classification scenarios. This work serves as the first systematic framework for multimodal class incremental learning, establishing a foundation that future research can build upon.

Our contributions are threefold: (1) We propose the first benchmark specifically designed for multimodal class-incremental learning, enabling standardized evaluation and comparison of models. (2) We adapt a state-of-the-art vision-language model to various continual learning strategies, shedding light on the strengths and limitations of these methods in a multimodal context. (3) We provide comprehensive experimental results that reveal key insights and identify promising directions for future research in the field. By addressing the need for a standardized evaluation protocol, our benchmark aims to catalyze research in multimodal continual learning, fostering a deeper understanding of how to effectively maintain knowledge across evolving multimodal data distributions.

## 2 RELATED WORK

### 2.1 INCREMENTAL LEARNING

Incremental learning addresses the challenge of continuously learning new information from dynamic, changing data streams (van de Ven et al., 2022; Mai et al., 2022; Qu et al., 2021). The problem can be framed using various scenarios, such as task-incremental or class-incremental learning, depending on how task identifiers are provided over time, with these being the most common settings considered in the literature (Wang et al., 2024). Various strategies tackle these challenges by enabling learning new tasks while mitigating the forgetting of previously acquired knowledge. For instance, regularization-based techniques apply constraints to specific parameters related to earlier tasks, thereby preserving prior knowledge and preventing catastrophic forgetting (Kirkpatrick et al., 2017; Li & Hoiem, 2017; Zenke et al., 2017). Replay-based methods leverage stored samples by maintaining data from previously learned tasks in a rehearsal buffer and continuously interleaving it with the training of new tasks, allowing for ongoing consolidation of past knowledge (Rolnick et al., 2019; Isele & Cosgun, 2018; Wang et al., 2024; Buzzega et al., 2020). More recently, efficient prompt-based rehearsal-free methods have emerged, combining powerful pretrained backbones with learnable prompts. This approach preserves knowledge across tasks without altering the backbone weights, thereby significantly mitigating forgetting in the entire system (Wang et al., 2022b;c; Smith et al., 2022; Razdaibiedina et al., 2023).

### 2.2 VISION-LANGUAGE MODELING

Vision-language modeling is at the intersection of computer vision and natural language processing. It seeks to develop models capable of understanding and generating multimodal information, where visual inputs are paired with corresponding linguistic descriptions. Transformer models significantly improved multimodal learning thanks to the inherent capability of self-attention operations to connect multimodal signals (Nagrani et al., 2021), and to the introduction of self-supervised pretaining paradigms specifically designed to perform joint representational learning (Singh et al., 2022; Bao et al., 2022; Wang et al., 2022a). The fusion of information from different modalities has been modeled adopting different strategies, ranging among early-, mid-, and late-fusion, based on the information processing stage where the two modalities are combined (Nagrani et al., 2021). These advancements enable robust multimodal alignment by establishing deep relationships and semantic correspondences between sub-components of visual and linguistic instances. Such capability allows vision-language models to succeed in many scenarios such as vision-language reasoning (Antol et al., 2015; Suhr et al., 2018; Goyal et al., 2017), text generation for image captioning (Chen et al., 2015), and text-to-image retrieval (Plummer et al., 2015; Lin et al., 2014). Furthermore, the easy access to foundational pretrained vision-language models, like CLIP (Radford et al., 2021), popularized few-shot classification via image-text contrastive fine-tuning (Zhou et al., 2022b;a; Khattak et al., 2023).

## 2.3 MULTIMODAL CONTINUAL LEARNING

Applications of continual learning strategies in multimodal settings, especially vision-language ones, are sparse and heterogeneous. A popular approach is to leverage the few-shot capabilities of vision-language pretrained models to turn a vision class-incremental learning problem into a multimodal one, where classification is achieved via contrastive learning (D'Alessandro et al., 2023; Yu et al., 2024; Thengane et al., 2022). These approaches build upon the strength of vision-language in leveraging text embeddings to represent visual classes incrementally. This strategy offers an implicit way to handle new visual categories by extending textual prompts with new concepts, although the task here is still inherently unimodal. Another approach is focused on task-incremental learning. In this framework, different tasks are learned continuously, where the system is not only concerned with the representational learning of new categories but also handling completely new tasks (Srinivasan et al., 2022) . This scenario seeks to mitigate catastrophic forgetting across diverse tasks by retaining knowledge from previously learned tasks, rather than solely focusing on previous data samples. Furthermore, a more established multimodal continual learning benchmark is found in VQA (Zhang et al., 2023; Kane et al., 2022; Zhang et al., 2022), where a system must answer natural language questions about images. The continual learning challenge in VQA revolves around the ability to adapt to new visual scenes and linguistic expressions over time, as the model is incrementally exposed to new domains with novel semantics, vocabulary, and visual environments. We observe that none of these approaches address the need for a benchmark specifically designed for pure classification tasks in multimodal scenarios.

## 3 MCIL BENCHMARK

In this section, we present the main components of the MCIL benchmark, namely, the datasets that constitute the classification challenge, the problem formulation, and the set of continual learning strategies adapted to vision-language modeling providing a conceptual baseline for the MCIL challenge.

### 3.1 DATASETS

The proposed benchmark comprises 3 datasets from distinct semantic domains. In each dataset, examples are presented as paired images and text that share a common semantic grounding. By ensuring this alignment between modalities, we emphasize the true strength of a multimodal model, where both visual and textual information must be jointly leveraged for effective classification.

**Caltech-UCSD Birds (CUB).** This dataset is built upon the original CUB database (Welinder et al., 2010) with the addition of model-based captions entailing fine-grained visual descriptions of bird images (Reed et al., 2016). The dataset contains 11.764 RGB images and 200 well-balanced classes representing bird species. Captions provide feature-by-feature rich structured information focusing on body part attributes, rather than generic informal visual descriptions. Paired samples where textual description might cue the bird species (e.g. the name of the species is contained in the caption) have been removed.

**Oxford Flowers.** This dataset is built upon the original Oxford Flowers database (Nilsback & Zisserman, 2008) with additional fine-grained visual descriptions as image captions (Reed et al., 2016). The dataset contains 8.189 RGB images and 102 unbalanced classes representing flower species. As for the previous dataset, textual descriptions provide structured information about body attributes. Paired samples where textual description might cue the flower species have been removed.

**DVM-CAR.** This dataset consists of a great database of car models built for marketing research purposes (Huang et al., 2022). The dataset is already multimodal since it aligns car images with a set of tabular features that cover various meta-data variables and form a relational database. The dataset contains more than 1 million samples and 286 heavily unbalanced classes representing car models. We applied a data transformation procedure to turn 13 variables in the table into a sentence containing the variable names and their respective value in a semantically consistent structure (e.g. "the [fuel type] is [diesel]", where the square brackets contain the original tabular feature name-value pair). Such sentences are paired with respective car images to gather the image-text paired sample.

## 3.2 PROBLEM FORMULATION

In the MCIL setting, we are given a stream of labeled training sets or experiences $E_1, E_2, \ldots, E_T$, where each experience $E_t = \{(\boldsymbol{x}_{i,t}, y_{i,t})\}_{i=1}^{N_t^E}$, consists of $N_t^E$ training examples. As in standard continual learning, classes do not overlap among experiences. In our setting, $\boldsymbol{x} = [v, l]$ is the multimodal sample with $v$ and $l$ corresponding to the aligned vision and language data, respectively. For any given experience $t$, classification takes place via the model:

$$g_{t,\Theta}(f_{t,\Phi}([v_t, l_t])) \tag{1}$$

where $f_{t,\Phi}(\cdot)$ is a multimodal feature extractor parameterized by $\Phi$, and $g_{t,\Theta}$ is a proper classification head parameterized by $\Theta$, $t = 1, 2, \ldots, T$.

**Evaluation Metric.** For the evaluation phase, experience-wise performance is computed by considering all the classes encountered up to the current experience $t$. Consider the stream of labelled evaluation sets $D_1, D_2, \ldots, D_T$, and model $g(f(\cdot))$, then the evaluation accuracy for experience $t$ is computed as follows:

$$A_t = \frac{\sum\limits_{(\boldsymbol{x}_i, y_i) \in D_1 \cup D_2 \cup \ldots D_t} [g_t(f_t(\boldsymbol{x}_i = [v_i, l_i])) = y_i]}{N_1^D + N_2^D + \ldots + N_t^D} \tag{2}$$

where $N_t^D$ is the number of evaluation examples for experience $t$, and $[\cdot]$ the indicator function. More precisely, score $A_t$ is the balanced accuracy defined as the accuracy score with class-balanced sample weights (Brodersen et al., 2010), to account for the unbalance of the selected benchmark datasets.

## 3.3 MODELS

To evaluate the behavior of vision-language models on the MCIL benchmark datasets, any model compatible with eq. 1 can be considered—specifically, models capable of producing a joint representation of vision and language instances that can later be used by a classification head. In general, most multimodal fusion paradigms satisfy this requirement.

However, most do not directly yield a compressed multimodal representation; instead, they rely on additional steps to combine the unimodal representations through mathematical operations such as concatenation, averaging, or summation of the unimodal hidden states. While these approaches ensure multimodal alignment, they require additional processing. Models like CLIP (Radford et al., 2021) and ALIGN (Jia et al., 2021) produce aligned vision and language hidden states separately, necessitating further adaptation to obtain a final multimodal representation that is compatible with eq. 1.

In contrast, models such as ViLBERT (Lu et al., 2019) and ViLT (Kim et al., 2021) inherently perform information fusion through cross-modal attention, although their outputs remain modality-specific. While these models are generally well-suited for a wide range of use cases and continual learning strategy adaptations, they may require additional engineering for specific methods—such as modern rehearsal-free approaches—that demand a single multimodal hidden state extracted from a frozen backbone. To address this issue, we aim to propose an unbiased vision-language adaptation that does not involve additional engineering or specialized architectural modifications.

Models such as VL-BEiT (Bao et al., 2022), BEiT3 (Wang et al., 2022a), and Flava (Singh et al., 2022), are ideal candidates, as they generate a joint multimodal embedding that represents both modalities during pretraining, without post hoc manipulation of aligned unimodal representations. Among these, we selected Flava as the primary model for evaluating vision-language continual learning adaptations on the MCIL benchmark, as it offers a balanced trade-off between complexity and flexibility while satisfying the constraints of eq. 1. Flava is a hierarchical model where data is first processed through separate, specialized vision and language encoders before being fed into a multimodal encoder, which performs attention-based fusion on the resulting unimodal hidden states.

A multimodal class token is also prepended to the joint hidden state sequence, producing the final multimodal representation used for classification.

To further enhance the proposed benchmark, we have tested Flava across various continual learning strategies, providing a comprehensive view of its performance under different scenarios.

**Upper Bound (UB).** The pretrained Flava model is fine-tuned on all the training sets of all the experiences up to the current experience.

**Lower Bound (LB).** The pretrained Flava model is continuously trained on every subsequent experience. In the first experience, the pretrained model is fine-tuned on the first experience, and the weights are left to update for subsequent experiences training sets.

**Dual Prompt (DP).** Pretrained parameters of the Flava model are kept frozen, while two types of learnable prompts are responsible for learning through experience and preserving previous knowledge (Wang et al., 2022b). In particular, E-Prompts and G-Prompts are responsible for learning task-invariant and task-specific knowledge, respectively. A different set of E-Prompts is learned for each experience via prefix-tuning, while G-Prompts are continuously updated through experiences to represent general knowledge across tasks. Both prompts are mounted on the two unimodal encoders, as well as on the multimodal encoder, independently. In this way, experience-wise model adaptation is modality-specific to account for unimodal data distribution shift, but also multimodal to account for joint, abstract distribution shift. During the evaluation phase, a query function is used to select the proper E-Prompts for a given example, from a prompts pool. The query function consists of learning a mapping between E-Prompts keys and class tokens obtained from a pretrained frozen static Flava model.

**Dual Prompt closed-form (DPcf).** This is a variant of the Dual Prompt model, where the mapping between the class token of a given example and the E-Prompts keys is not learned, but it is computed analytically by solving the optimization problem in closed-form (Appendix A for details). E-Prompts and G-Prompts are applied and free to be learned as in standard Dual Prompt, but the E-Prompts keys no longer need to be learned since they are computed analytically for each experience, independently.

**Learning to Prompt (L2P).** The Flava model is kept frozen and prompt-tuning is applied to learn a mapping between experience samples and a preferred set of prompts for that experience (Wang et al., 2022c). A set of top-N prompts are extracted from a pool of prompts, and prepended to the hidden states via concatenation before passing to the encoder. The top-N prompts are selected via a similarity score between the class token of a given example and prompt keys, in order to assign a likelihood to the most suitable prompts for that experience. As in Dual Prompt, the class token for computing the similarity score is obtained by extracting the multimodal class token of the data sample from a frozen static Flava model.

**Experience Replay (ER).** The pretrained Flava model is fine-tuned on the first experience training set, and trained on subsequent experience with the aid of a sample buffer containing image-text pairs from past experiences. We considered two versions of Experience Replay, one where the buffer is iteratively filled with a subsample of $25\%$ of examples per class of the previous experience, namely, ER25, and one with a $10\%$ subsampling rate, namely, ER10.

## 4 EXPERIMENT

### 4.1 IMPLEMENTATION DETAILS

We use the Flava model and its corresponding pretrained weights from the HuggingFace Transformers library (Wolf et al., 2020) as our initial backbone. All models and experimental pipelines are implemented using a custom PyTorch package with support from the Avalanche library (Lomonaco et al., 2021). For all continual learning strategies, the training is conducted for 5 epochs using AdamW optimizer (Loshchilov, 2017). The learning rate is set to $0.005$ for L2P, DP, and DPcf, and $1e-5$ for all other strategies. The batch size is uniformly set to 16 for both training and evaluation across all datasets and models.

All experiments are executed on three GeForce RTX 3090 Ti GPUs. To ensure robust model comparison, we report the average evaluation accuracy, as defined in eq. 2, computed over 3 indepen-

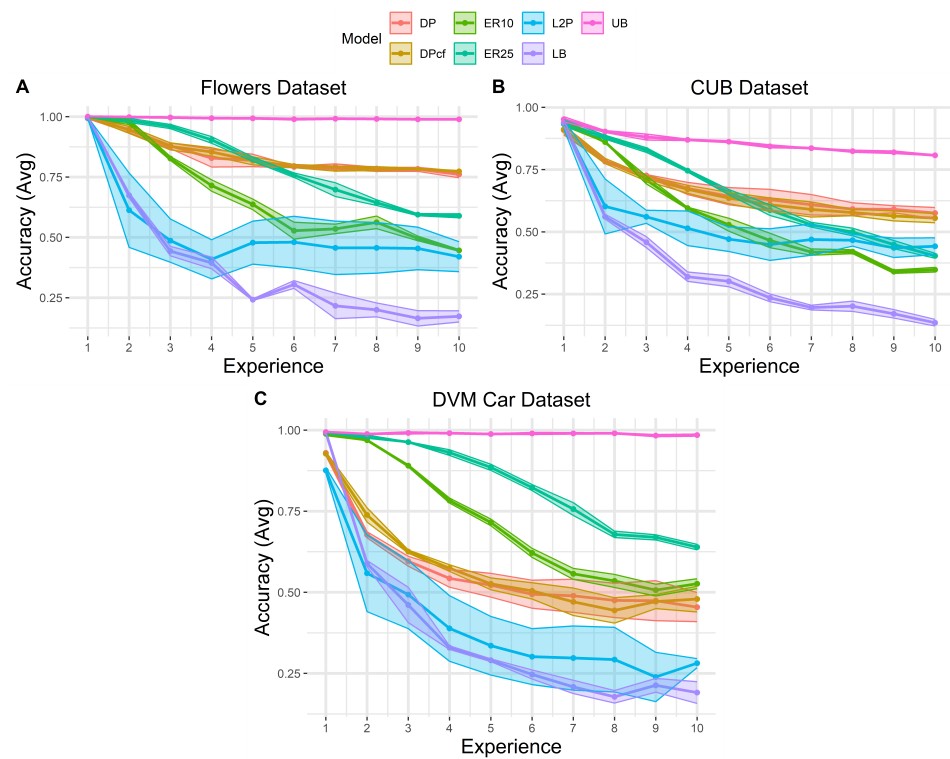

Figure 1: Results.

Table 1: Average balanced accuracy across all experiences (standard errors in parenthesis).

| | **Dataset** | | |
|---|---|---|---|
| **Model** | **CUB** | **Flowers** | **DVM Car** |
| DP | $67.30 \pm 3.33$ | $83.77 \pm 2.46$ | $56.54 \pm 4.58$ |
| DPcf | $66.13 \pm 3.61$ | $84.14 \pm 2.45$ | $57.62 \pm 4.82$ |
| ER25 | $65.20 \pm 5.96$ | $79.54 \pm 5.09$ | $83.17 \pm 4.34$ |
| ER10 | $56.23 \pm 6.66$ | $67.22 \pm 6.33$ | $70.93 \pm 5.96$ |
| L2P | $53.42 \pm 4.75$ | $52.49 \pm 5.51$ | $40.63 \pm 6.11$ |
| LB | $35.17 \pm 7.77$ | $38.11 \pm 8.48$ | $36.99 \pm 8.08$ |
| UB | $85.97 \pm 1.39$ | $99.31 \pm 0.12$ | $98.90 \pm 0.10$ |

dent runs. The datasets are split into 10 incremental experiences, each containing an equal number of classes. However, the total number of samples and class-specific distributions may vary across experiences.

## 4.2 RESULTS

In Table 1, we present the evaluation results of the vision-language continual learning adaptations to the benchmark datasets, as the average across-experience balanced accuracy (Figure 1). A complete table of results showing evaluation performance for every experience is presented in Appendix B.

The experimental results indicate that the upper bound performance is notably high, with minimal accuracy decay across experiences, suggesting that the classification task itself is not inherently complex. However, all tested continual learning methods experience substantial performance degradation and forgetting, highlighting the difficulty of the setting due to shifts in multimodal data distribution. The gap between the upper bound and the results achieved by continual learning strategies underscores the challenge of balancing stability and plasticity over time.

Moreover, the sensitivity of various datasets to different families of methods varies considerably. Specifically, parameter-efficient approaches that utilize multimodal pretraining knowledge by keeping the backbone frozen tend to exhibit better long-term performance on the CUB and Flowers datasets. Conversely, the DVM Car dataset demonstrates the opposite pattern, with these strategies underperforming, emphasizing the need for more specialized adaptation techniques that are able to efficiently handle multimodal data distributions.

This discrepancy arises from the nature of the image-text relationship in the datasets. In DVM Car, the language data is represented as sentence-formatted tabular information, which lacks the semantic richness found in the textual captions of the CUB and Flowers datasets. As a result, prompt-based methods, which rely on adapting a language backbone, face additional challenges when the semantic connection between image and text is weak. In this context, experience replay methods have an inherent advantage, unless further prompt engineering or multimodal adaptations are employed.

The continual learning adaptation of L2P demonstrates suboptimal performance overall, contrasting with its success in vision-only continual learning tasks (Wang et al., 2022c). A key observation is that DPcf performs comparably to its non-analytical, and original, variant. DP-like methods, which utilize a query function to match a class token with a learnable experience-related key, can be computationally expensive, as they require an additional forward pass to extract the class token from a frozen pretrained backbone. This computational cost becomes particularly burdensome in multimodal scenarios, where models tend to be larger and more resource-intensive than in unimodal tasks. However, DPcf reduces this overhead by requiring the additional forward pass only once per experience, rather than at every training step. This provides a computational advantage, especially when dealing with complex models like Flava, which have hierarchical architectures.

This efficiency is one reason why multimodal models that construct a multimodal class token during pretraining have a natural advantage in parameter-efficient continual learning. By building a unified representation for vision-language fusion, such models streamline the learning process and reduce computational costs, making them well-suited for multimodal continual learning scenarios.

## 5 CONCLUSIONS

In this paper, we introduced the MCIL benchmark, the first multimodal continual learning benchmark designed specifically for evaluating multimodal continual learning methods in classification tasks. Using the Flava architecture as a baseline, we evaluated how vision-language models adapt to incremental learning scenarios. Our experimental results reveal that the proposed datasets pose varying challenges to different methods, largely due to differences in the inherent semantic alignment between image and text instances.

The benchmark also underscores the key challenges of adapting multimodal models to continual learning scenarios. While vision-language models provide strong representations, their adaptation to continual learning is complex, particularly in handling multimodal data and shifting distributions. In this study, we applied basic adaptation schemes of existing continual learning strategies to maintain consistency with the original methods. However, these often resulted in suboptimal performance, suggesting that more advanced approaches are required to fully harness the potential of these models in class-incremental classification tasks.

Future research should focus on more sophisticated prompt engineering, improved multimodal integration techniques, and parameter-efficient adaptations to enhance both robustness and scalability in continual learning environments. The proposed benchmark datasets can serve as a common foundation for testing such advancements.

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

## A  ANALYTICAL SOLUTION OF THE E-PROMPTS KEY OPTIMIZATION IN THE DUAL PROMPT MODEL

The objective function for DP (Wang et al., 2022b) includes a query-key matching term in addition to the standard cross-entropy loss, where the query is the function extracting the class token of a sample from a static frozen backbone, and the key is a learnable embedding. The goal of the optimization problem is to find the key embedding satisfying the following expression:

$$\max_k \sum_{i=1}^{N} \gamma(q(x_i), k) = \max_k \sum_{i=1}^{N} \frac{q(x_i) \cdot k}{\|q(x_i)\|\|k\|} \tag{3}$$

where $\gamma$ is the cosine similarity function, $q(x_i)$ the query function for the *i-th* sample, $k$ a learnable key, $q(x_i) \cdot k$ indicates the dot product, and $N$ the number of samples in the experience training set. In the original paper, $k$ is optimized via back-propagation. However, we observe that $k$ has an analytical solution. Let's normalize the query vector by defining:

$$\tilde{q}_i = \frac{q(x_i)}{\|q(x_i)\|}. \tag{4}$$

Substituting this into the objective function gives:

$$\max_k \sum_{i=1}^{N} \tilde{q}_i \cdot k =$$

$$\max_k \left( \sum_{i=1}^{N} \tilde{q}_i \right) \cdot k \tag{5}$$

where the second step is guaranteed by the linearity of the dot product. Note that we omit $\|k\|$ if we assume $k$ has a unit form already, that is $\|k\| = 1$. Let $Q$ represent the sum of the normalized query vectors:

$$Q = \sum_{i=1}^{N} \tilde{q}_i.$$

The objective function now becomes:

$$\max_{k} \ Q \cdot k. \tag{6}$$

Since $Q$ needs to be a unit vector for the cosine similarity operation, maximization is ensured when $k$ is computed as follows:

$$k = \frac{Q}{\|Q\|}$$
$$= \frac{\sum_{i=1}^{N} \frac{q(x_i)}{\|q(x_i)\|}}{\left\|\sum_{i=1}^{N} \frac{q(x_i)}{\|q(x_i)\|}\right\|}. \tag{7}$$

The optimal key can be computed with a single forward pass through all the examples of any given experience training set.

# B   FULL RESULTS

The following tables report the dropping rate (PD) metric, which measures the drop in accuracy in the last experience w.r.t. the accuracy in the first one as a measure of forgetting, and the across-experience average balanced accuracy as a measure of overall performance.

Table 2: CUB200 results.

| Method | Avg Accuracy in each session (%) | | | | | | | | | | Avg. ↑ | PD ↓ |
| | 1 | 2 | 3 | 4 | 5 | 6 | 7 | 8 | 9 | 10 | | |
|---|---|---|---|---|---|---|---|---|---|---|---|---|
| LB | 93.80 | 56.11 | 45.87 | 31.99 | 30.14 | 23.49 | 19.60 | 20.12 | 17.09 | 13.52 | 35.17 | 58.63 |
| L2P | 93.30 | 60.19 | 56.04 | 51.44 | 47.05 | 44.88 | 46.90 | 46.63 | 43.59 | 44.22 | 53.42 | 39.87 |
| ER10 | 93.84 | 86.04 | 70.83 | 59.53 | 52.85 | 46.53 | 41.86 | 42.04 | 34.01 | 34.78 | 56.23 | 37.61 |
| ER25 | 93.84 | 88.00 | 82.84 | 74.58 | 65.96 | 58.72 | 52.85 | 49.95 | 44.87 | 40.40 | 65.20 | 28.64 |
| DPcf | 91.15 | 78.46 | 71.49 | 67.11 | 63.56 | 60.88 | 58.99 | 57.76 | 56.39 | 55.49 | 66.13 | 25.02 |
| DP | 90.82 | 78.26 | 72.62 | 67.42 | 64.32 | 62.90 | 60.87 | 59.13 | 59.07 | 57.54 | 67.30 | 23.52 |
| UB | 95.18 | 90.28 | 88.06 | 86.95 | 86.21 | 84.34 | 83.57 | 82.37 | 81.96 | 80.75 | 85.97 | 9.21 |

Table 3: Flowers results.

| Method | Avg Accuracy in each session (%) | | | | | | | | | | Avg. ↑ | PD ↓ |
| | 1 | 2 | 3 | 4 | 5 | 6 | 7 | 8 | 9 | 10 | | |
|---|---|---|---|---|---|---|---|---|---|---|---|---|
| LB | 99.90 | 67.42 | 44.31 | 39.44 | 24.19 | 30.51 | 21.64 | 19.96 | 16.47 | 17.26 | 38.11 | 61.79 |
| L2P | 99.53 | 61.25 | 48.66 | 40.90 | 47.80 | 48.01 | 45.64 | 45.66 | 45.40 | 42.02 | 52.49 | 47.05 |
| ER10 | 99.90 | 97.75 | 82.68 | 71.48 | 63.67 | 52.78 | 53.53 | 56.24 | 49.57 | 44.60 | 67.22 | 32.68 |
| ER25 | 99.90 | 98.44 | 95.93 | 90.38 | 82.39 | 75.78 | 69.79 | 64.40 | 59.40 | 58.94 | 79.54 | 20.37 |
| DP | 99.31 | 94.73 | 87.96 | 82.84 | 81.89 | 79.20 | 79.28 | 78.21 | 78.17 | 76.08 | 83.77 | 15.55 |
| DPcf | 99.85 | 94.52 | 87.69 | 85.42 | 81.73 | 79.56 | 78.50 | 78.49 | 78.33 | 77.34 | 84.14 | 15.71 |
| UB | 100.0 | 99.79 | 99.68 | 99.37 | 99.32 | 98.94 | 99.16 | 99.06 | 98.89 | 98.91 | 99.31 | 0.69 |

Table 4: DVM cars results.

| Method | Avg Accuracy in each session (%) | | | | | | | | | | Avg. ↑ | PD ↓ |
| | 1 | 2 | 3 | 4 | 5 | 6 | 7 | 8 | 9 | 10 | | |
|---|---|---|---|---|---|---|---|---|---|---|---|---|
| LB | 99.33 | 58.87 | 46.09 | 32.94 | 29.05 | 24.66 | 20.83 | 17.77 | 21.31 | 19.09 | 36.99 | 62.34 |
| L2P | 87.59 | 55.88 | 49.30 | 38.85 | 33.53 | 30.16 | 29.73 | 29.24 | 23.90 | 28.15 | 40.63 | 46.96 |
| DP | 92.99 | 67.74 | 59.58 | 54.29 | 52.19 | 49.39 | 48.92 | 47.51 | 47.41 | 45.43 | 56.54 | 36.44 |
| DPcf | 92.74 | 73.85 | 62.54 | 57.46 | 52.62 | 50.43 | 47.08 | 44.40 | 47.15 | 47.93 | 57.62 | 35.12 |
| ER10 | 98.86 | 96.90 | 89.02 | 78.28 | 71.55 | 62.15 | 55.68 | 53.51 | 50.67 | 52.64 | 70.93 | 27.93 |
| ER25 | 99.13 | 97.99 | 96.28 | 93.08 | 88.53 | 82.28 | 75.67 | 67.86 | 66.94 | 63.94 | 83.17 | 15.96 |
| UB | 99.39 | 98.81 | 99.13 | 99.07 | 98.81 | 98.97 | 99.00 | 99.00 | 98.31 | 98.47 | 98.90 | 0.49 |

