# OpenReview forum: "A Multimodal Class-Incremental Learning benchmark for classification tasks"
_ICLR.cc/2025/Conference — ICLR 2025 Conference Withdrawn Submission_

### Official Review · Reviewer_JZ9f · 2024-10-27

**Soundness:** 1
**Presentation:** 3
**Contribution:** 1
**Rating:** 1
**Confidence:** 4

**Summary:**

The paper presents a new benchmark for multimodal class-incremental learning (MCIL) with a focus on multimodal classification tasks. The benchmark is formed using three existing benchmarks that consist of images along with their captions.  5 CL methods along with an upperbound and a lowerbound using Flava as the base model are tested on the benchmark by splitting each dataset into 10 incremental tasks.

**Strengths:**

I hardly can find a strengths in this work. The only thing that comes to my mind is that the writing quality is acceptable and the paper can be followed straightforwardly.

**Weaknesses:**

The contribution of this work is extremely weak. Three existing benchmarks are split and then five CL methods are run on the resulting tasks. Hence, new data is collected nor a new method is introduced. The paper is not well-motivated and it is not clear what this paper is going to offer to the research community.

**Questions:**

The paper is so weak and naive that it is not easy to come up with questions. The reason is that a work should be novel enough such that one can ask meaningful questions about the approach. I hardly could come up with the following questions:


1. What unique challenges does multimodal class-incremental learning present that are not captured by unimodal benchmarks? Other than changing the model from unimodal to multimodal, what is the specific challenge that makes this dataset specific?

2. What is the novelty and motivation of this work? What specific gaps in existing multimodal continual learning research does this benchmark address?

3. How this work can help researchers? How does this benchmark compare to or improve upon existing multimodal or continual learning datasets? How might researchers use this benchmark to develop new multimodal continual learning algorithms?

---

### Official Review · Reviewer_1dMk · 2024-10-29

**Soundness:** 2
**Presentation:** 2
**Contribution:** 2
**Rating:** 3
**Confidence:** 4

**Summary:**

The paper proposes a novel multimodal Class-incremental learning benchmark to standardize the evaluation of VLM models in a continual learning setting. It evaluates a widely used VLM with multiple existing continual learning methods on three curated multimodal datasets. The work is focused on multimodal classification tasks keeping other variables like tasks and domains fixed.

**Strengths:**

It is a focused work on the classification task while keeping other factors fixed. Such a controlled study can lead to more precise conclusions about the strengths and challenges of different methods in visual-language settings. The discussion of different methods is quite detailed. However, it should be supported with ablations and quantitative analysis.

**Weaknesses:**

1. The VLM methods and datasets included in this work only cover a narrow domain. A much wider study is required for establishing such a benchmark.  The work generalizes the observation using different continual learning methods using only one VLM model. More VLM models (at least one from each type) from Section 3.3 like CLIP or ALIGN, ViLBERT and latest approach like BLIP should be included in the evaluation to make stronger conclusions.

2. The datasets covered in this work belong to very specific domains like birds, cars, and flowers. More generic datasets should be included to make it a comprehensive framework.

3. The setting of class-incremental learning studied in this work is not realistic because the continuous data usually does not arrive in pure chunks of classes and is often a mix of all classes in different proportions. For example, use something like 'the CLEAR Benchmark' by Lin et al. It captures a realistic temporal evolution of visual concepts.

4. The limit of the paper is 10 pages. Additional results should in included in the main paper instead of the Appendix to improve the presentation of the paper. Details about the Dual Prompt (DP) model can be included in the main manuscript.

5. The details about experiment settings including model implementation details are missing. For example - Was the Flava model pretrained on some dataset or was it trained from scratch? The reported standard errors are calculated on how many runs? How big is the training, validation and test set for each dataset?

6. Presentation can be improved. Captions should be self-complete containing information about the experiment setup and conclusions. Captions like ‘Results.’ are not acceptable.

**Questions:**

1. The presentation of the paper should be improved including captions, figures and experiment details.
2. Add more VLM models to make conclusions more robust to the model choice.
3. I would recommend adding more ablations and quantitative analysis to make reasoning about the performance of continual learning methods stronger.
4.  Also evaluate on generic datasets to make it a comprehensive framework.

Overall, I would recommend authors add more methods and generic datasets, provide more analysis, and pick a more realistic continual learning setting to set up a valuable multimodal CIL benchmark for the community.

---

### Official Review · Reviewer_S92B · 2024-10-30

**Soundness:** 2
**Presentation:** 1
**Contribution:** 1
**Rating:** 1
**Confidence:** 4

**Summary:**

This paper addresses the gap in multimodal continual learning by introducing a novel benchmark for Multimodal Class-Incremental Learning (MCIL), designed specifically for classification tasks across multiple modalities. While continual learning has progressed in the vision and language domains independently, multimodal continual learning remains underexplored, particularly in classification contexts. Current research has focused on specific applications like VQA and text-to-vision retrieval, lacking a standardized benchmark to fairly compare and evaluate methods in multimodal settings.

The proposed MCIL benchmark includes a curated selection of multimodal datasets for classification and adapts a popular vision-language model (Flava) across various continual learning strategies. This setup allows a systematic evaluation of model performance in handling incremental learning with diverse data sources. By providing experimental insights into the behavior of vision-language models within this framework, the paper establishes a foundation for future research in multimodal continual learning, aiming to enhance knowledge retention and cross-modal alignment over time.

**Strengths:**

This paper gives lots of description about the used models, metrics and datasets, clearly illustrating the pipeline and experimental settings.

**Weaknesses:**

1. The motivation for this work is unclear. Given the strong capabilities of current vision-language models, why is incremental learning for multimodal classification necessary? Presently, vision-language models can handle image classification effectively with either visual or text input alone; both modalities are not needed simultaneously to perform well. Additionally, I noticed that in the proposed MCIL benchmark, the authors remove samples where the description explicitly mentions the ground-truth category. However, this approach does not fully prevent leakage, as the text may still indirectly reveal category information through similar words or phrases that are difficult to detect and remove. It would be helpful if the authors could clarify the specific use case or scenario for this task to help readers better understand the motivation behind it.

2. The presentation is poor and the experiments are insufficient. It seems a semi-finished product submission with just 7 pages of main contents.

**Questions:**

See weaknesses.

---

> ### Author Response · Authors · 2024-11-13
>
> We thank the reviewer for the comments. We addressed the point as follows.
> (1) We kinda disagree with the view that MCIL is not necessary just because vision-language models are powerful. As shown in the MCIL benchmark and model comparison, the challenge exists since performance drops substantially during experiences. Regardless of the intrinsic power of current vision-language models, you can't have a clue of an existing conceptual/theoretical/practical problem until you don't test the models against new challenges. Furthermore, multimodal vision-language classification is an industry-relevant problem, although is not popular in academic scenarios, it's one of the most requested use cases in some industry niche scenarios, and currently, there is not a single insight in the literature regarding how to handle pure multimodal classification problems in continual learning. We also think the argument that both modalities are not needed simultaneously to perform well should be formally demonstrated, not just stated. And maybe here we should have demonstrated this point. This is something we can do, and we thank the reviewer for pointing out this issue. Regarding the issue of removing samples containing the ground truth class label in the text part, we ended up removing just a few examples per dataset, in the order of 1-10 samples, so a negligible portion of the data. No concern for leakage.
> (2) We agree that the presentation is not rich enough, and incomplete. We can improve a lot here.

---

### Official Review · Reviewer_Ny1G · 2024-10-31

**Soundness:** 2
**Presentation:** 1
**Contribution:** 2
**Rating:** 3
**Confidence:** 4

**Summary:**

This paper presents the Multimodal Class-Incremental Learning (MCIL) benchmark, aimed at evaluating multimodal continual learning methods specifically within classification tasks. The authors utilize the Flava architecture as a baseline to assess how vision-language models adapt to incremental learning scenarios. They identify significant challenges faced by these models when dealing with multimodal data and shifting distributions.

**Strengths:**

- The majority of existing research focus on single modality based continual learning.The introduction of the MCIL benchmark is important in the field of multimodal continual learning. By providing a standardized framework, it fills a critical gap in current research, enabling more consistent evaluation and comparison of models.

- The paper includes some experimental results of current methodologies in Multimodal Class-Incremental Learning.

**Weaknesses:**

- While the paper acknowledges that current methods yield suboptimal performance, it does not sufficiently explore or propose more advanced method that could better leverage the potential of multimodal information. A deeper discussion on potential techniques for multimodal incremental learning is important.
- The writing is unclear and the paper is less organized. For example, it is unclearer how to adapt existing continual learning strategies for multimodal incremental learning. Providing more detailed explanations would improve the reproducibility of the experiments.
- The paper uses Flava architecture as a baseline. However, CLIP is more popular in vision and language model, and the experiments based on CLIP is missing.
4.As a benchmark, there are only three datasets in the incremental process, which is not enough for reflecting the ability of incremental learning. Besides, more experimental results and analysis are needed.

**Questions:**

Weakness Section

---

### Official Review · Reviewer_U3Tg · 2024-11-02

**Soundness:** 1
**Presentation:** 2
**Contribution:** 2
**Rating:** 3
**Confidence:** 5

**Summary:**

This manuscript introduces a benchmark for Multimodal Class-Incremental Learning (MCIL). The benchmark comprises a selection of multimodal datasets tailored to classification challenges.

**Strengths:**

This paper proposed a multimodal CIL benchmark, and migrate some classical methods to this benchmark for testing. Continual learning techniques are deployed on a vision-language model.

**Weaknesses:**

Weakness：
1: The test datasets of the benchmark are few and simple, which are only three common datasets.
It doesn't look like a complete and good benchmark.

2: There are already multimodal CIL protocols in existence, such as [1], [2], [3], and your benchmark seems to overlap with them to a high degree. This benchmark seems to be not significant.
[1]: CLIP MODEL IS AN EFFICIENT CONTINUAL LEARNER
[2]: AttriCLIP: A Non-Incremental Learner for Incremental Knowledge Learning
[3]: Class-Incremental Learning with CLIP: Adaptive Representation Adjustment and Parameter Fusion

3: The experimental method for migration is not sufficient. The Dual-Prompt and L2P methods used are relatively classic but outdated methods. The latest methods such as [4], [5], [6], [7], etc. may also need to be migrated to further measure the necessity of the benchmark.

[4]: Coda-prompt: Continual decomposed attention based prompting for rehearsal-free continual learning
[5]: Promptfusion: Decoupling stability and plasticity for continual learning
[6]: One-stage Prompt-based Continual Learning
[7]: Hierarchical decomposition of prompt-based continual learning: Rethinking obscured sub-optimality


4: The multimodal model selected in this paper is also relatively outdated. If more advanced MLLM models such as LLava1.5, CogVLM, CogVLM2 are used to conduct experiments on the benchmark, the significance of the benchmark can be further explained.

5: The content of the paper is insufficient and lacks sufficient analysis and explanation of the particularity and necessity of this benchmark.

**Questions:**

see the weakness

---

> ### Author Response · Authors · 2024-11-13
>
> We thank the reviewer for the comments. We addressed the points as follows:
> (1) we agree that the benchmark is limited. However, we made an effort to select the best pure vision-language candidates for multimodal classification. Finding relevant pure multimodal datasets is not easy since they are quite unpopular, and we did our best to at least provide the basis for a new challenge.
> (2) This issue has been raised by a previous reviewer, so I'll re-post the answer here. We think there is a misunderstanding. The cited papers are irrelevant to the current proposal. The mentioned papers are meant for unimodal scenarios. The classification problem is treated in a multimodal fashion, but as very standard for CLIP-based few-shot classification, the true modality is the vision one. The language modality is just a textual rearrangement of the classification label. So you use unimodal datasets. This is not the context in which we are proposing our benchmark. We are proposing a multimodal classification benchmark. That means that the dataset already contains image and text data aligned, both carrying their pack of semantic knowledge (this is different from engineering the class label of an unimodal dataset into a sentence of the form "a photo of a [class]").
> (3) we agree that the methods proposed for model comparison are not comprehensive. We can extend the baseline model comparison with new methods and work on further Flava adaptation. We thank the reviewer for pointing out the new method to include.
> (4) The purpose of this paper was to propose a new benchmark, to to obtain a new sota. We selected a representative model designed for pure multimodal problems. We think this provides a conceptually relevant baseline to understand the challenge of multimodal CIL, more relevant than picking the newest models for the newest sota since new models and sota are produced every week.
> (5) We agree that the paper lacks a deep explanation and justification in most sections. We believe the paper can improve a lot from this perspective.

---

### Official Review · Reviewer_wN1q · 2024-11-03

**Soundness:** 1
**Presentation:** 2
**Contribution:** 1
**Rating:** 3
**Confidence:** 5

**Summary:**

This paper introduces a multimodal class-incremental learning benchmark, consisting of Oxford Flowers, CUB and DVM-CAR datasets. The authors select Flava as the primary model for evaluating vision-language continual learning on the MCIL benchmark. The authors also include 3 baselines (DualPrompt, L2P, Experience Replay) for multimodal class-incremental learning evaluation.

**Strengths:**

This paper introduces a Multimodal Class-Incremental Learning benchmark. The paper demonstrates the relevance and utility of the proposed benchmark by adapting a widely used model (Flava) and providing insights into its performance.

**Weaknesses:**

1. My major concern is that the contribution is limited. This benchmark is referred to as multimodal, but it only includes visual and language modalities, and is limited to image classification tasks. However, such a so-called multimodal class-incremental learning (MCIL) classification task has already been investigated. There are some works on class-incremental learning using visual-language models[1-5] and it seems that there is no difference between their setting and the proposed MCIL setting. The authors should clarify and carefully discuss the differences between them. Besides, apart from the visual and language modalities, there are also works investigating the audio-visual incremental learning problem [6], and I suggest the authors should conduct a comprehensive literature review before proposing a new benchmark. The authors should explicitly compare their proposed benchmark to the existing works, highlighting any key differences or improvements. Additionally, the authors should expand their literature review to include audio-visual and other multimodal incremental learning approaches, and discuss how their benchmark relates to or differs from these broader multimodal settings.
2. The datasets CUB, Oxford Flowers, and DVM-CAR are all small-scale datasets. Some core settings are missing from the experiments, such as experiments on ImageNet1K and experiments with different numbers of tasks, which are widely adopted in different class-incremental learning works. I suggest the authors to justify their choice of datasets and explain why they believe these are sufficient for evaluating multimodal class-incremental learning.
3. I cannot understand why methods similar to CLIP and ALIGN do not meet Eq1. Although these methods cannot generate a unified multimodal feature representation, they still accomplish classification tasks by calculating cosine similarities. At the same time, I also fail to understand how Flava is used. What is the text input of Flava in the MCIL setting?
4. The survey of related work is not comprehensive enough, and the compared baseline methods are not comprehensive. The recent works only include L2P and DualPrompt, which is insufficient for a benchmark evaluation. Since the authors choose Flava, which generates a unified feature representation, classic CIL methods such as Regularization-based methods (e.g. LwF) and replay-based methods (e.g. iCaRL) can also be used in the Flava model. I suggest the authors could refer to [7] for a comprehensive review of CIL methods. The authors are advised to supplement relevant experiments.
5. Some minor issues:
    - The number of images in line 144 and line 152, '.'->','
    - The title of Figure 1 is too simple.
    - In Table 1, it would be easier to understand if UB and LB are listed separately, and the best results among other methods are bolded. The formatting of Table 1 could also be improved.
    - I suggest the authors add a figure to illustrate the whole pipeline of the proposed MCIL setting.

[1] Select and Distill: Selective Dual-Teacher Knowledge Transfer for Continual Learning on Vision-Language Models (ECCV24)

[2] Boosting Continual Learning of Vision-Language Models via Mixture-of-Experts Adapters (CVPR2024)

[3] Continual Vision-Language Representation Learning with Off-Diagonal Information (ICML2023)

[4] Preventing Zero-Shot Transfer Degradation in Continual Learning of Vision-Language Models (ICCV 2023)

[5] Generative Negative Text Replay for Continual Vision-Language Pretraining (ECCV2022)

[6] Audio-visual class-incremental learning. (ICCV23)

[7] A Comprehensive Survey of Continual Learning: Theory, Method and Application (TPAMI 2024)

**Questions:**

1. What is the text input of the Flava model in the MCIL setting?
2. What is the difference between the proposed MCIL and existing CLIP-based CIL problem?
3. Why Oxford Flowers, CUB and DVM-CAR are chosen, why not include larger-scale dataset?

---

> ### Author Response · Authors · 2024-11-13
>
> We thank the reviewer for the time dedicated to reading and commenting on the paper. We addressed the points as follows:
> (1) We think there is a big misunderstanding since the cited papers are irrelevant to the current proposal. Except for the Audio-Visual CIL work, the mentioned papers are meant for unimodal scenarios. The classification problem is treated in a multimodal fashion, but as very standard for CLIP-based few-shot classification (in continual learning), the true modality is the vision one. The language modality is just a textual rearrangement of the classification label. So you use unimodal datasets. This is not the context we are proposing our benchmark in. We are proposing a multimodal classification benchmark. That means that the dataset already contains image and text data aligned, both carrying their pack of semantic knowledge (this is different from engineering the class label of a unimodal dataset into a sentence of the form "a photo of a [class]". We agree that the audio-related literature should be included in the references. A pure Vision-Language classification benchmark still doesn't exist in the literature.
> (2) ImageNet1K is a vision dataset. How does it relate to our multimodal vision-language scenario? We can maybe better clarify in the main manuscript why there is a crucial difference between unimodal and vision-language datasets and why we selected pure vision-language datasets instead of vision-only ones. However, CUB, Flowers, and DVM cars, as stated in the manuscript, are a set of curated datasets ready for multimodal classification. We discarded several irrelevant datasets, but we are open to advice regarding how to extend the benchmark.
> (3) CLIP and ALIGN accomplish classification tasks in unimodal scenarios, not in multimodal ones. In multimodal scenarios, both information (vision and language) carry information for the final classification. CLIP-related methods are unbalanced since the true data is the image, and the text data is ad-hoc engineered text reflecting the class label. To turn CLIP into a pure multimodal classifier you have to pick the final vision embedding and the final text embedding and engineer them to map onto classification nodes. The need to have this additional step made us prefer Flava-like models, intrinsically designed to have an end-to-end multimodal classification mechanism. I hope this difference is clear. We can clarify this in the manuscript.
> (4) We agree that more methods should be included, and as suggested by the reviewer, most methods like regularization-based ones are compatible with Flava. So we can include more methods in the model comparison.
>
>
> Questions.
> (1) The text input is the sentence associated with an image in the original multimodal datasets.
> (2) MCIL is for multimodal classification, CLIP-based CIL is for unimodal classification.
> (3) Those 3 datasets are the only ones that survived a long selection process. Industry-relevant, pure vision-language datasets, are unpopular, and most of them are too trivial.

---

### Note · Authors · 2024-11-18

I have read and agree with the venue's withdrawal policy on behalf of myself and my co-authors.